# Country of birth, educational level and other predictors of seeking care due to decreased fetal movements: an observational study in Sweden using data from a cluster-randomised controlled trial

Ingela Radestad,[1] Karin Pettersson,[2] Helena Lindgren,[3] Viktor Skokic,[4] Anna Akselsson [5]

For numbered affiliations see end of article.

**Correspondence to**
Dr Anna Akselsson;
anna.akselsson@shh.se

## ABSTRACT

**Objectives** To identify predictors of seeking care for decreased fetal movements and assess whether care-seeking behaviour is influenced by Mindfetalness.

**Design** Observational study with data from a cluster-randomised controlled trial.

**Setting** 67 maternity clinics and 6 obstetrical clinics in Sweden.

**Participants** All pregnant women with a singleton pregnancy who contacted the obstetrical clinic due to decreased fetal movements from 32 weeks' gestation of 39 865 women.

**Methods** Data were collected from a cluster-randomised controlled trial where maternity clinics were randomised to Mindfetalness or routine care. Mindfetalness is a self-assessment method for women to use daily to become familiar with the unborn baby's fetal movement pattern.

**Outcome measures** Predictors for contacting healthcare due to decreased fetal movements.

**Results** Overall, 5.2% (n=2059) of women contacted healthcare due to decreased fetal movements, among which 1287 women (62.5%) were registered at a maternity clinic randomised to Mindfetalness and 772 women (37.5%) were randomised to routine care. Predictors for contacting healthcare due to decreased fetal movements were age, country of birth, educational level, parity, prolonged pregnancy and previous psychiatric care (p<0.001). The main differences were seen among women born in Africa as compared with Swedish-born women (2% vs 6%, relative risk (RR) 0.34, 95% CI 0.25 to 0.44) and among women with low educational level compared with women with university-level education (2% vs 5.4%, RR 0.36, 95% CI 0.19 to 0.62). Introducing Mindfetalness in maternity care increased the number of women seeking care due to decreased fetal movements overall.

**Conclusion** Women with country of birth outside Sweden and low educational level sought care for decreased fetal movements to a lesser extent compared with women born in Sweden and those with university degrees. Future research could explore whether pregnancy outcomes can be improved by motivating women in these groups to contact healthcare if they feel a decreased strength or frequency of fetal movements.

### Strengths and limitations of this study

► This is the first study exploring the association between country of birth and contact due to decreased fetal movements.
► The new information about predictors for contacting healthcare due to decreased fetal movements can be valuable for preventing stillbirth.
► Zero lost to follow-up, large sample size and use of a population-based register are methodological strengths.
► Misclassified subjects might have affected the results for the predictors and it is probably an underestimation of the number of women seeking care.
► Possible confounders, not being considered in the study, could have had an impact on the result.

**Trial registration number** NCT02865759.

## INTRODUCTION

A woman's interpretation of signals from the fetus and her ability to take action will determine whether she contacts healthcare when she has concerns for her unborn baby's health. Her response to altered fetal movement patterns can be decisive for the pregnancy outcomes. Among women who have sought care due to decreased fetal movements, reports indicate that one-quarter will later experience adverse baby outcomes, such as low Apgar score, small for gestational age or stillbirth.[1–3] Further, up to 50% of women who experience stillbirth report a gradual decrease in fetal movements several days before the baby died.[4 5] The majority waited more than 24 hours before contacting healthcare without the perception of any movements.[6 7] Contact with healthcare due to decreased fetal movements is common and most of the women return home after normal

findings are confirmed. The prevalence of women contacting healthcare with concerns of fetal movements varies internationally; from 6% to 34%.[8–11]

Women with normal pregnancies can distinguish between many different types of fetal movements and most feel strong movements.[12] Maternal perception of fetal movements and becoming familiar with the movement pattern is important in reducing stillbirth, as reported in research from New Zealand.[13] The character and diurnal rhythm of movements seem to be essential factors in preventing stillbirth. Maternally perceived increased strength of fetal movements is associated with reduced stillbirth rate (adjusted odds ratio (aOR) 0.21, 95% CI 0.12 to 0.36) and quiet or light movements in the evening were associated with higher risk (aOR 3.82, 95% CI 1.57 to 9.31).

Raising maternal awareness of fetal movements by using the Mindfetalness method may be a way forward for women to become familiar with the unborn baby's movement pattern.[14] Compared with counting methods,[15 16] Mindfetalness instruct the women to observe the variation of fetal movements. Practicing the method includes to lie down for 15 min a day from 28 weeks' gestation, when the baby is awake and to focus on the character, strength and frequency of the unborn baby's movements, without counting each movement. Mindfetalness was invented in 2012[14] and has been evaluated by women and midwives in two studies.[17 18] Additionally, in a cluster-randomised controlled trial, 67 maternity clinics were allocated to either Mindfetalness or routine care.[19 20] Midwives in the Mindfetalness group were instructed to distribute oral and written information about fetal movements and about Mindfetalness. Information regarding the method was available in nine different languages and a website was open for anyone to access. The trial showed several benefits of introducing Mindfetalness into maternity care.[19] In the Mindfetalness group, more women started their labour spontaneously, followed by a reduction in labour inductions and caesarean sections. Further, less babies were born small for gestational age and needed neonatal care.

The ambition to encourage women to seek care if they have concerns of their unborn baby's fetal movements pattern is to reduce the stillbirth rate without too much overload on both women and caregivers. However, stillbirth is a very rare phenomenon and thus comparing stillbirth rate in this study is impossible. In this study, we investigate predictors of seeking care for decreased fetal movements and whether care-seeking behaviour was influenced by a method (Mindfetalness) aimed at increasing pregnant women's knowledge of their unborn baby's fetal movement patterns.[14] We further investigate pregnancy outcomes for women seeking care for decreased fetal movements but where, at the time of the examination, no interventions such as induction of labour or caesarean section were made.

## METHODS

Pregnant women with a singleton pregnancy who contacted healthcare due to decreased fetal movements from 32 weeks' gestation constitute the population of this study. Data were collected from a cluster-randomised controlled trial in which 67 maternity clinics in Stockholm, Sweden, were randomised to either intervention with Mindfetalness or routine care, as reported previously.[19 20] The women who contacted healthcare due to decreased fetal movements from 32 weeks' gestation were generated from the International Classification of Diseases (ICD) code AM041 (examination due to decreased fetal movements).[21] The code is a diagnosis for pregnant women who have contacted healthcare due to decreased fetal movements, where no interventions, such as labour induction or caesarean section, were made, at the time the woman sought care.

The randomisation resulted in 33 maternity clinics in the Mindfetalness group and 34 maternity clinics in routine care group. After the randomisation, one clinic randomised to Mindfetalness declined participation. We applied intention-to-treat design; extended information about the randomisation process can be found in previous publications.[19 20] The Mindfetalness group included 19 639 women and the routine care group included 20 226 women.

Starting in October 2016, midwives, working at maternity clinics randomised to the intervention, distributed a leaflet to pregnant women, including information about fetal movements and the self-assessment method, Mindfetalness. The last leaflet was distributed on 31 January 2018, and the women were followed-up until birth in population-based registers. Midwives in the routine care group followed standard care, which includes providing oral information about fetal movements at 24 weeks' gestation.

The Mindfetalness leaflet (online supplemental file 3) included information about fetal movements and when to contact healthcare; 'If you are concerned that the fetus is moving less or that the movements are weaker, you should contact healthcare'. Further, the leaflet included an invitation to practice Mindfetalness daily from 28 weeks' gestation. The information was available in nine different languages and the website www.mindfetalness.com was open for anyone to access.

## STATISTICAL ANALYSIS

We identified the background factors of the women who contacted healthcare due to decreased fetal movements by calculating percentages. The included variables were age, country of birth, education level, parity, assisted reproduction, previous stillbirth, tobacco use at registration in maternity clinic, body mass index and maternal diseases. To calculate predictors for contacting healthcare, we investigated which factors were statistically significantly associated with contact due to decreased fetal movements. As a metric for association, we calculated

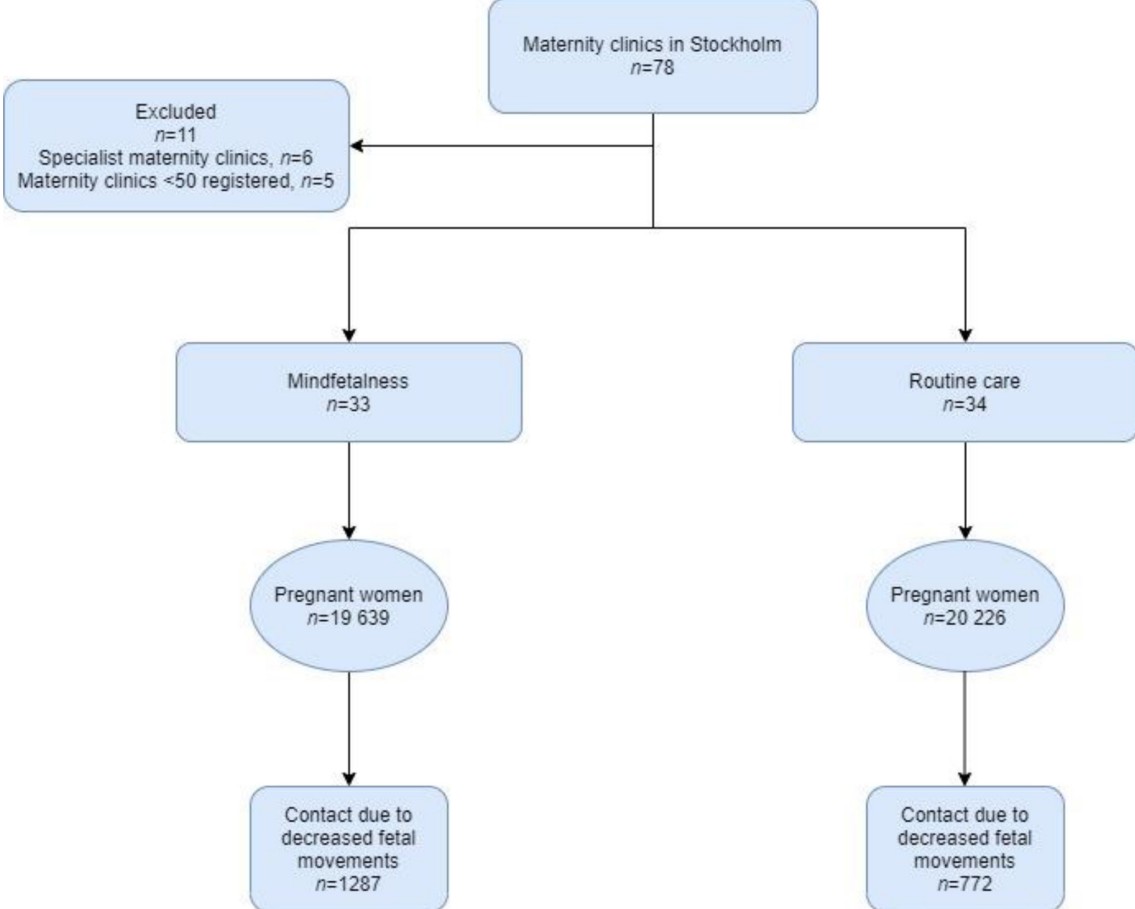

**Figure 1** Flow chart.

## RESULTS

Among all women in this study, 5.2% (n=2059) contacted healthcare due to decreased fetal movements, measured from gestational week 32 until birth. As shown in figure 1, 62.5% (n=1287) were registered at a maternity clinic randomised to Mindfetalness, and 37.5% (n=772) to routine care. The background data for the women are displayed in table 1.

Predictors for seeking care for decreased fetal movements were age, country of birth, educational level, parity, prolonged pregnancy and previous psychiatric care (p<0.001) (table 2). The main differences were seen in country of birth and educational levels. Women born in Africa sought care to a lesser extent as compared with women born in Sweden, 2% vs 6% (RR 0.34, 95% CI 0.25 to 0.44).

Further, women with lowest educational levels sought care due to decreased fetal movements to a lesser extent compared with women with university-level education (2% vs 5.4%, RR 0.36, 95% CI 0.19 to 0.62). Additionally, primiparas, women with prolonged pregnancies (>41+6 gestation weeks), and women with experience of psychiatric care more often sought care due to decreased fetal movements (table 2). The predictors for contacting healthcare due to decreased fetal movements remain after adjustment for parity. Mindfetalness increased the prevalence of women contacting healthcare due to decreased fetal movements overall (6.6% vs 3.8%, RR 1.72, 95% CI 1.57 to 1.87) (not in table). When stratifying women into groups according to background factors, the percentage of women contacting healthcare increased in all groups (online supplemental table 1).

Figure 2A,B shows the pregnancy outcomes for women contacting healthcare due to decreased fetal movements, stratified for women registered at maternity clinics randomised to Mindfetalness and routine care, respectively. Compared with routine care, women in the Mindfetalness group had a higher percentage of women with spontaneous start of labour (66.4% vs 63%) and a lower rate of labour induction (25.6% vs 28.6%). The

relative risks (RRs) and 95% CIs using log-binomial regression. In the predictor calculation, we adjusted for the possible confounding factor, parity. For outcome measures, we calculated percentage and compared the Mindfetalness group with the routine care group using log-binomial regression. We considered a p value of 0.05 to be statistically significant. We used the statistical programme R (V.3.2.4). Loss to follow-up was zero due to the population-based Swedish Pregnancy Register[22] used for data collection.

**Table 1** Characteristics among women with a singleton pregnancy from 32 weeks' gestation contacting healthcare versus not contacting healthcare due to decreased fetal movements

| Characteristics | Contacted healthcare due to decreased fetal movements (n=2059) n (%) | Did not contact healthcare due to decreased fetal movements (n=37 806) n (%) | Relative risk (CI) | P value |
|---|---|---|---|---|
| Age* | | | | |
| ≤24 years | 187 (9.1) | 2628 (7) | 1.31 (1.13 to 1.51) | <0.001 |
| 25–34 years | 1361 (66.1) | 24 137 (63.8) | 1.04 (1 to 1.07) | 0.04 |
| ≥35 years | 511 (24.8) | 11 041 (29.2) | 0.85 (0.79 to 0.92) | <0.001 |
| Country of birth | | | | |
| Sweden | 1595 (77.5) | 24 890 (65.8) | 1.18 (1.15 to 1.21) | <0.001 |
| Europe (except Sweden) | 137 (6.7) | 3780 (10) | 0.67 (0.56 to 0.79) | <0.001 |
| Africa | 51 (2.5) | 2450 (6.5) | 0.38 (0.29 to 0.50) | <0.001 |
| Asia | 224 (10.9) | 5663 (15) | 0.73 (0.64 to 0.82) | <0.001 |
| North America | 16 (0.8) | 304 (0.8) | 0.97 (0.59 to 1.60) | 1 |
| South America | 31 (1.5) | 622 (1.6) | 0.92 (0.64 to 1.31) | 0.72 |
| Others | 5 (0.2) | 97 (0.3) | 0.95 (0.39 to 2.33) | 1 |
| Education level† | | | | |
| Shorter than 9 years | 11 (0.5) | 551 (1.5) | 0.36 (0.20 to 0.66) | <0.001 |
| Elementary school | 69 (3.4) | 1734 (4.6) | 0.72 (0.57 to 0.91) | 0.006 |
| High school | 546 (26.5) | 9717 (25.7) | 1.02 (0.95 to 1.10) | 0.64 |
| University | 1290 (62.7) | 22 714 (60.1) | 1.03 (1 to 1.06) | 0.09 |
| Parity‡ | | | | |
| Primipara | 1127 (54.7) | 16 344 (43.2) | 1.26 (1.21 to 1.32) | <0.001 |
| Multipara | 930 (45.2) | 21 324 (56.4) | 0.80 (0.76 to 0.84) | <0.001 |
| Assisted reproduction‡ | 129 (6.3) | 2033 (5.4) | 1.17 (0.98 to 1.38) | 0.09 |
| Previous stillbirth | 9 (0.4) | 211 (0.6) | 0.78 (0.40 to 1.52) | 0.64 |
| Tobacco user at registration in the maternity clinic (including snuff)‡ | 79 (3.8) | 1256 (3.3) | 1.16 (0.92 to 1.44) | 0.21 |
| Body mass index§ | | | | |
| <18.5 kg/m² | 54 (2.6) | 1021 (2.7) | 0.97 (0.74 to 1.27) | 0.89 |
| 18.5–24.9 kg/m² | 1228 (59.6) | 22 624 (59.8) | 0.99 (0.96 to 1.03) | 0.70 |
| 25–29.9 kg/m² | 464 (22.5) | 8611 (22.8) | 0.99 (0.91 to 1.07) | 0.74 |
| 30–34.9 kg/m² | 167 (8.1) | 2889 (7.6) | 1.06 (0.91 to 1.23) | 0.47 |
| ≥35 kg/m² | 63 (3.1) | 1000 (2.6) | 1.15 (0.90 to 1.48) | 0.26 |
| Maternal diseases | | | | |
| Diabetes mellitus | 1 (0) | 63 (0.2) | 0.29 (0.04 to 2.10) | 0.26 |
| Coronary heart disease | 25 (1.2) | 600 (1.6) | 0.77 (0.51 to 1.14) | 0.20 |
| Thrombosis | 13 (0.6) | 283 (0.7) | 0.84 (0.49 to 1.47) | 0.69 |
| Systemic lupus erythematosus | 1 (0) | 58 (0.2) | 0.32 (0.04 to 2.28) | 0.37 |
| Psychiatric care | 354 (17.2) | 4744 (12.5) | 1.37 (1.24 to 1.51) | <0.001 |
| Endocrine disease | 156 (7.6) | 2633 (7) | 1.09 (0.93 to 1.27) | 0.29 |
| Epilepsy | 10 (0.5) | 167 (0.4) | 1.10 (0.58 to 2.08) | 0.73 |
| Chronic hypertension | 9 (0.4) | 184 (0.5) | 0.90 (0.46 to 1.75) | 0.87 |
| Other disease | 204 (9.9) | 3583 (9.5) | 1.05 (0.91 to 1.20) | 0.51 |

Continued

**Table 1** Continued

| Characteristics | Contacted healthcare due to decreased fetal movements (n=2059) n (%) | Did not contact healthcare due to decreased fetal movements (n=37 806) n (%) | Relative risk (CI) | P value |
|---|---|---|---|---|
| Medication or psychological treatment for mental illness | 129 (6.3) | 2058 (5.4) | 1.15 (0.97 to 1.37) | 0.11 |

*Mean age of women who contacted care=31.2, who did not contact care=32.4 (p<0.001).
†Data missing for women who contacted care n=143 (6.9%), who did not contact care n=3090 (8.2%).
‡Data missing for women who contacted care n=2 (0.1%), who did not contact care n=138 (0.4%).
§Data missing for women who contacted care n=83 (4%), who did not contact care n=1661 (4.4%).

proportion of women who had onset of labour from 42 weeks' gestation was 6.8% in the Mindfetalness group, and 7.8% in routine care. Further, a lower proportion of babies in the Mindfetalness group were in need of transfer to neonatal intensive care unit (6.3% vs 8%). One stillbirth was observed, in the routine care group (not in table). There were no statistically significant differences in the investigated pregnancy outcomes in figure 2A,B, between the Mindfetalness and routine care groups. For further information in pregnancy outcomes for Mindfetalness and routine care, see online supplemental table 2.

## DISCUSSION

During the studied period, in the Stockholm region, women born outside Sweden and women with low levels of education sought care for decreased fetal movements to a lower extent when compared with women born in Sweden and women with university-level education. Further, women of higher age contacted healthcare due to decreased fetal movements to a lower extent than women of a younger age. Country of birth outside Sweden, low educational level and advanced maternal age have previously been related to adverse birth outcomes. Introducing Mindfetalness in maternity care increased the number of women contacting healthcare due to decreased fetal movements.

We found that women born outside Sweden (mostly from Africa and Asia) sought care due to decreased fetal movements to a lower extent. No studies, to our knowledge, have investigated the association between contact with healthcare due to maternal concern of fetal movements and country of birth. However, previous research has reported a higher risk of stillbirth among women not being born, but giving birth, in a Western country.[23–28] Suggested mechanisms for this difference include substandard maternity care, lack of communication between the pregnant woman and healthcare professionals and variations in pre-pregnancy health.[24 29] It is possible that the higher risk for stillbirth can be linked to lower awareness and knowledge about fetal movements and when to contact healthcare.[6 7]

Giving information to pregnant women about fetal movements may reduce pre-hospital delays[30–32] and has been suggested to reduce the stillbirth rate in an intervention study performed in Norway.[32] The study documented a 50% reduction in the stillbirth rate among women who contacted healthcare due to decreased fetal movements as compared with women who did not. The intervention included providing health promotion information to the pregnant women, an invitation to count fetal movements, and information that they should never wait until the next day to contact healthcare if they had concerns about fetal movements. Additionally, providing new guidelines to the healthcare professionals was included in the intervention. A reduction in patient delay and increased knowledge about fetal movements was shown in a recent study from the Netherlands after the introduction of a leaflet about fetal movements.[33] However, in the AFFIRM trial,[34] women in the intervention arm (including information to pregnant women and new guidelines to the healthcare professionals) had lower rate of stillbirths than women in the control group, but the difference was not statistically significant. Labour inductions and caesarean section increased, and the number of babies in need of neonatal care more than 48 hours.

The midwife is the most important source of information[35 36] and The Swedish National Board of Health and Welfare recently introduced new guidelines for maternity care, including instructions to midwives to inform pregnant women about fetal movements.[37] The guidelines have good compliance among midwives (79%–87% answered yes to the different questions whether they followed the guidelines), but some report barriers due to lack of time, high workload and problems in communicating with women born outside of Sweden.[38] Suboptimal care is more common among non-western women and action to improve perinatal mortality for this group is needed.[24]

In this study, women with low education levels sought care due to decreased fetal movements to a lower extent, as compared with women with high education levels. A higher risk for stillbirth has been found among women with low educational levels and low socioeconomic status.[6 39–41] There are reasons to believe that raising awareness about fetal movements among women with low education or socioeconomic status would increase the

**Table 2** Predictors for contacting healthcare due to decreased fetal movements

| Predictor | P value | n (%) | RR (CI) | P value | Adjusted RR (CI)* |
|---|---|---|---|---|---|
| Age | <0.001 | | | 0.02 | |
| ≤24 years | | 187 (6.6) | 1.24 (1.07 to 1.44) | | 1.15 (0.98 to 1.33) |
| 25–34 years | | 1361 (5.3) | 1 (Reference) | | 1 (Reference) |
| ≥35 years | | 511 (4.4) | 0.83 (0.75 to 0.91) | | 0.90 (0.82 to 1) |
| Country of birth | <0.001 | | | <0.001 | |
| Sweden | | 1595 (6) | 1 (Reference) | | 1 (Reference) |
| Europe (except Sweden) | | 137 (3.5) | 0.58 (0.49 to 0.69) | | 0.58 (0.49 to 0.69) |
| Africa | | 51 (2) | 0.34 (0.25 to 0.44) | | 0.36 (0.27 to 0.47) |
| Asia | | 224 (3.8) | 0.63 (0.55 to 0.72) | | 0.66 (0.57 to 0.75) |
| North America | | 16 (5) | 0.83 (0.49 to 1.29) | | 0.83 (0.49 to 1.28) |
| South America | | 31 (4.7) | 0.79 (0.54 to 1.09) | | 0.82 (0.56 to 1.13) |
| Others | | 5 (4.9) | 0.81 (0.30 to 1.70) | | 0.84 (0.31 to 1.75) |
| Marital status | 0.20 | | | 0.06 | |
| Cohabiting with becoming father | | 1914 (5.2) | 1 (Reference) | | 1 (Reference) |
| Single | | 29 (4) | 0.76 (0.52 to 1.07) | | 0.73 (0.49 to 1.02) |
| Other family situation | | 114 (4.8) | 0.92 (0.76 to 1.10) | | 0.87 (0.72 to 1.04) |
| Education level | <0.001 | | | <0.001 | |
| Shorter than 9 years | | 11 (2) | 0.36 (0.19 to 0.62) | | 0.41 (0.21 to 0.69) |
| Elementary school | | 69 (3.8) | 0.71 (0.56 to 0.90) | | 0.75 (0.58 to 0.94) |
| High school | | 546 (5.3) | 0.99 (0.90 to 1.09) | | 1 (0.91 to 1.10) |
| University | | 1290 (5.4) | 1 (Reference) | | 1 (Reference) |
| Parity | <0.001 | | | NA | |
| Primipara | | 1127 (6.5) | 1.54 (1.42 to 1.68) | | NA |
| Multipara | | 930 (4.2) | 1 (Reference) | | NA |
| Gestation week >41+6 | <0.001 | | | 0.01 | |
| No | | | 1 (Reference) | | 1 (Reference) |
| Yes | | | 1.34 (1.13 to 1.56) | | 1.25 (1.06 to 1.46) |
| Assisted reproduction | 0.10 | | | 0.55 | |
| No | | 1928 (5.1) | 1 (Reference) | | 1 (Reference) |
| Yes | | 129 (6) | 1.16 (0.97 to 1.38) | | 1.05 (0.88 to 1.25) |
| Previous stillbirth | 0.46 | | | 0.95 | |
| No | | 2050 (5.2) | 1 (Reference) | | 1 (Reference) |
| Yes | | 9 (4.1) | 0.79 (0.38 to 1.41) | | 0.98 (0.47 to 1.74) |
| Tobacco user at registration at the maternity clinic | 0.22 | | | 0.17 | |
| No | | | 1 (Reference) | | 1 (Reference) |
| Yes | | 79 (5.9) | 1.15 (0.92 to 1.42) | | 1.17 (0.93 to 1.44) |
| Body mass index | 0.77 | | | 0.37 | |
| <18.5 kg/m² | | 54 (5) | 0.98 (0.74 to 1.26) | | 0.95 (0.72 to 1.23) |
| 18.5–24.9 kg/m² | | 1228 (5.1) | 1 (Reference) | | 1 (Reference) |
| 25–29.9 kg/m² | | 464 (5.1) | 0.99 (0.89 to 1.10) | | 1.03 (0.92 to 1.14) |
| 30–34.9 kg/m² | | 167 (5.5) | 1.06 (0.90 to 1.24) | | 1.12 (0.95 to 1.31) |
| ≥35 kg/m² | | 63 (5.9) | 1.15 (0.89 to 1.46) | | 1.22 (0.94 to 1.54) |
| Psychiatric care | <0.001 | | | <0.001 | |
| No | | 1705 (4.9) | 1 (Reference) | | 1 (Reference) |

Continued

**Table 2** Continued

| Predictor | P value | n (%) | RR (CI) | P value | Adjusted RR (CI)* |
|---|---|---|---|---|---|
| Yes | | 354 (6.9) | 1.42 (1.27 to 1.58) | | 1.37 (1.22 to 1.52) |
| Medication or psychological treatment for mental illness | 0.12 | | | 0.13 | |
| No | | 1930 (5.1) | 1 (Reference) | | 1 (Reference) |
| Yes | | 129 (5.9) | 1.15 (0.96 to 1.36) | | 1.15 (0.96 to 1.36) |

*Adjusted for parity.
RR, relative risk.

number of women seeking care due to decreased fetal movements, which in turn would possibly decrease perinatal mortality.

Women of advanced maternal age contacted healthcare due to decreased fetal movements to a lower extent than younger women in our study, and a higher risk of stillbirth is reported among women aged from 35 years compared with younger women.[42] The mean age in an Israeli study was lower among women who had contacted

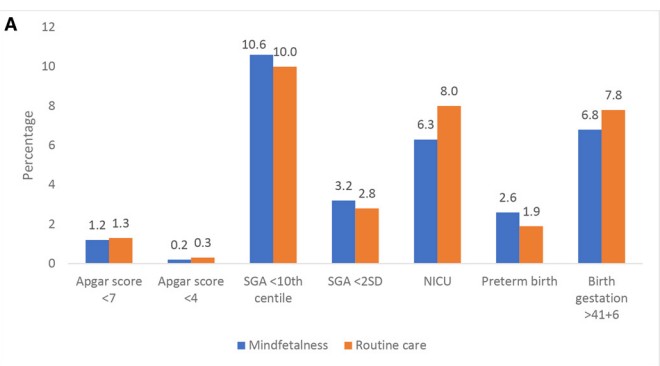

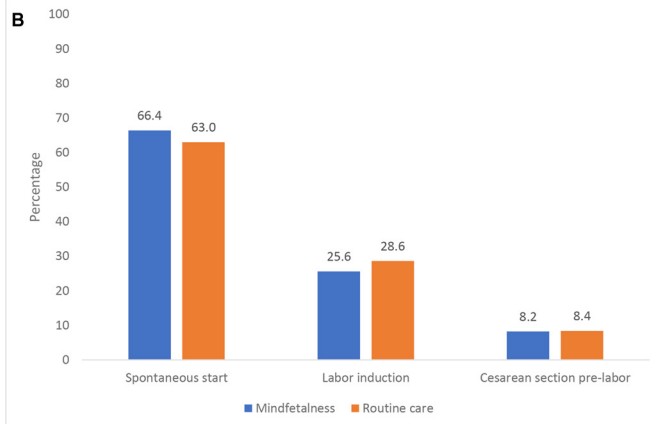

**Figure 2** (A) Birth outcome among women who contacted healthcare due to decreased fetal movements from 32 weeks' gestation, where the examination did not indicate that the pregnancy needed to be terminated, at the time the women sought care. (B) Start of labour among women who contacted healthcare due to decreased fetal movements from 32 weeks' gestation, where the examination did not indicate that the pregnancy needed to be terminated, at the time the women sought care. NICU, neonatal intensive care unit. SGA, small for gestational age.

healthcare due to decreased fetal movements compared with the reference group.[43] In a study from Norway, the percentage of women over 35 years of age was lower among women who contacted healthcare due to decreased fetal movements compared with the reference group (16% vs 20%).[3] However, in a recent systematic review, the authors interpreted current evidence as not supporting the notion that a woman's mean age influences whether she seeks care for decreased fetal movements.[44]

Introducing Mindfetalness in maternity care increased the number of women contacting healthcare due to decreased fetal movements overall. Mindfetalness possibly increases the woman's ability to become familiar with the unborn baby's fetal movement patterns. Reducing prehospital delay when the perception that fetal movements are reduced is suggested to be one way for improving perinatal mortality.[7 30 31]

## METHODOLOGICAL CONSIDERATIONS

The prevalence of women seeking care due to decreased or altered fetal movements in our study is low compared with the figures from other research. In a study performed in Sweden in 2014, 9.3% of the women sought care due to decreased fetal movements when measured from gestational week 28 until birth.[9] Further, international studies show a prevalence of 6%–34%.[8 10 11 45] We believe that the figures in our study are underestimated, mostly due to the fact that we only included women who sought care from gestational week 32. Additionally, the ICD code AM041 is not the best tool to use for including all women who contact healthcare due to decreased fetal movements, but it is the only code available. Misclassification in the study may be important to consider as some women who sought care due to decreased fetal movements did not get the ICD code and were classified as they had not sought care. This might have diluted the differences between the groups of women who sought/not sought care due to decreased fetal movements. However, when comparing the Mindfetalness group with routine care group, only non-differential misclassification is likely, which effect the differences in the same way in the two groups.

The only potential confounder we adjusted for was parity. We do not have information about other possibly confounders, for example, if some of the women had

extra appointments in specialist maternity clinics, which may have affected their care-seeking behaviour. Extra prebooked meetings with midwives or obstetricians and investigations with cardiotocography or ultrasound due to other complication can also decrease the need for unscheduled visits at the hospitals.

When evaluating whether Mindfetalness has an effect on women contacting healthcare due to decreased fetal movements, the random allocation, the large sample size and zero lost to follow-up are strengths of the study. Further, the missing data are negligible, thanks to the unique population-based register in Sweden. While the intention-to-treat design does preserve the advantages of random allocation, it also risks misclassification and the dilution of the results, that is, it is an underestimation of the true effects. From the pilot study, we know that approximately 75% of the Swedish-speaking women use the method,[17] and, from the main trial, about 22% did not receive a leaflet.[19] Additionally, the information was available in nine languages and, in Stockholm, Sweden, there is a large variety of women who originate from different countries and who speak languages that were not included in the intervention and this may have diluted the effect. This further contributes to the possible dilution of effects.

Less likely to seek care were (1) women 35 and over, (2) less educated and (3) non-Swedish women. Women using Mindfetalness were more likely to visit healthcare-providers due to decreased fetal movements than women in the control-group. It is unclear from this study whether we would have seen the same results if we had used a conventional 'count to ten' approach to maternal awareness of fetal movements

## IMPLICATIONS

The National Board of Health and Welfare claim that pregnant women should receive information about risk factors for adverse pregnancy outcomes, including stillbirth.[38] Women born in low-income countries and women with advanced maternal age are factors to consider when trying to improve perinatal mortality in high-income countries.[42 46] Providing targeted information, available in a variety of languages, to these women about fetal movements and when to contact healthcare might help to improve perinatal outcomes. Also, providing information about how to practice Mindfetalness can be one way forward to reduce adverse pregnancy outcomes. Today, we have a great number of opportunities to reach pregnant women thanks to the internet and to target women in at-risk groups. Additionally, this information can be provided in different forms, for example, videos, apps and interactive webpages.

## CONCLUSION

We do not know for certain what the ideal frequency is for women seeking care due to decreased fetal movements, but we can conclude that it is lower in women who have migrated to Sweden from, for example, Africa and Asia, and in women with low educational levels. Women from low-income countries have documented worse pregnancy outcomes. An implication of our results may be to examine whether pregnancy outcomes can be improved by providing customised information to women from low-income countries with the goal that they seek care if they feel a decreased strength or frequency of fetal movements. More data are needed to draw conclusions about the effects of Mindfetalness in perinatal mortality among women contacting healthcare due to decreased fetal movements.

**Author affiliations**
[1]Reproductive Health, Sophiahemmet University, Stockholm, Sweden
[2]Department of Clinical Science, Intervention and Technology, Karolinska Institute, Stockholm, Sweden
[3]Women's and Children's Health, Karolinska Institute, Stockholm, Sweden
[4]Institute of Clinical Sciences, Sahlgrenska Academy, University of Gothenburg, Goteborg, Sweden
[5]Health Promoting Science, Sophiahemmet University, Stockholm, Sweden

**Acknowledgements** The authors would like to thank the midwives and the pregnant women in the maternity clinics in Stockholm. Thanks to the coordination of midwives and medical doctors at Mödrahälsovårdsenheten and to The Swedish Pregnancy Register for cooperation. Special thanks to The Swedish Research Council for funding this study.

**Contributors** IR, KP, HL and AA were involved in the design and planning of the cluster-randomised controlled trial and IR obtained funding. AA had overall responsibility for the study. VS and AA analysed the data with input from IR, KP and HL. AA wrote the original manuscript and all authors made critical input. All authors were responsible for drafting and validating the following versions of the manuscript. All authors approved the final version of the manuscript.

**Funding** This study was funded by The Swedish Research Council. Grant number not applicable.

**Competing interests** None declared.

**Patient consent for publication** Not required.

**Ethics approval** This study was approved by the Regional Ethics committee in Stockholm, Sweden (Dnr 2015/2105–31/1), 13 January 2016.

**Provenance and peer review** Not commissioned; externally peer reviewed.

**Data availability statement** Data may be obtained from a third party and are not publicly available. No additional data is available. Anonymised data are available after ethical approval by the Regional Ethics committee in Stockholm, Sweden, and after request to The Swedish Pregnancy Register.

**ORCID iD**
Anna Akselsson http://orcid.org/0000-0003-0830-217X

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
