## [Reviewer comments · BMJ Open]

ARTICLE DETAILS

TITLE (PROVISIONAL)	Country of birth, educational level and other predictors of seeking care due to decreased fetal movements: An observational study in Sweden using data from a cluster-randomized controlled trial
AUTHORS	Radestad, Ingela; Pettersson, Karin; Lindgren, Helena; Skokic, Viktor; Akselsson, Anna

VERSION 1 – REVIEW

REVIEWER	Stacey, Tomasina University of Huddersfield, Department of Nursing and Midwifery
REVIEW RETURNED	21-Mar-2021

GENERAL COMMENTS	Thank you for giving me the opportunity to review this manuscript on the important issue of predictors for women contacting health services with a perception of reduced fetal movements. The research/paper is of relevance and is well written and think it will be of interest to researchers and service providers alike. I only have a couple of minor comments: 1. In the introduction, in line 37, the sentence starts 'thus' but there is not an obvious connection between it and the previous sentence (I suggest removing 'thus')2. I am aware this is a sub-analysis of the large 'Mindfetalness' and it is stated within the paper that there was an analysis between the % of women who accessed healthcare from the Mindfetalness group and routine care, and birth outcomes are presented between the 2 groups, however access to healthcare is not presented, although it is mentioned that Mindfetalness increased the access to care. I think it would be useful to present the differences in access to health care by background groups as this would help provide insight into whether Mindfetalness is equally effective amongst women with different socio-demographic characteristics, in particular for migrant women.
---

REVIEWER	Matsubara, Shigeki Jichi Medical School, Department of Obstetrics and Gynecology
REVIEW RETURNED	08-Apr-2021

GENERAL COMMENTS	To authors, The data is interesting and important. I have some advice to improve the paper quality. 1. Introduction: I believe that Mindfetalness is your own contrivance (procedures that your team established/proposed), right? Then, first, please state this meaning (you established the one). Second, shortly state what Mindfetalness is like: I believe that the followings (15 minutes, no counts, subjective self-evaluation, etc.) may be the keys of this procedure/methods.
--

	2. Introduction: I consider that you think that Mindfetalness is “better” than DFM-measurement hitherto reported (there have been many), and thus you did this study. Then, please shortly state in what point Mindfetalness is preferable (superior) to other methods previously reported. If this is one step to progress/widen Mindfetalness, then, “Mindfetalness is very good” is a very important message for the readers. 3. Roughly speaking, this study demonstrated TWO kinds of things at the same time. First, women using Mindfetalness is more likely to visit healthcare-providers than its counterparts. Second, what kind of women is less likely to visit them?: they were: i) aged women, ii) less educated, and iii) non-Swedish women. Then, the relation between the first and second, what? For example, of above three factors (less-likely-visit fraction), Mindfetalness affected what? I mean: “the data showed (or we believe) that Mindfetalness reduced (or may have reduced) the fraction of women of no-visit of i), or ii), or iii), what”. Its clarification may be useful to “improve” Mindfetalness system. If you have no data or no idea regarding this, please state this in the limitation section. I mean that “if you had used conventional count-ten”, i)ii)iii) may have become different. 4. The endpoint of DFM should be “to reduce the stillbirth without too much overload on both women and care-givers”. Stillbirth is a very rare phenomenon and thus comparing stillbirth rate in this study is impossible: I do understand. You state the meaning at the last of the text. However, since this journal is not OBYN specific, I believe that this should be definitely mentioned for the ordinary readers. This study did not show the very point; stillbirth reduction. 5. Also, for educational meaning, instead of count-to ten and other conventional methods (counting method), subjective movement count (like Mindfetalness) may be better to pick-up jeopardized fetus, if you consider so. 6. I made several advices and when you agree with me and decide to incorporate them, please do not expand the paper volume. In my opinion, the present paper is already too long to describe this simple data.
--	--

VERSION 1 – AUTHOR RESPONSE

Reviewer: 1

Dr. Tomasina Stacey, University of Huddersfield

Comments to the Author:

Thank you for giving me the opportunity to review this manuscript on the important issue of predictors for women contacting health services with a perception of reduced fetal movements. The research/paper is of relevance and is well written and think it will be of interest to researchers and service providers alike.

I only have a couple of minor comments:

1. In the introduction, in line 37, the sentence starts ‘thus’ but there is not an obvious connection between it and the previous sentence (I suggest removing ‘thus’)

To comply: We have deleted the word “thus” in page 3, line 91.

2. I am aware this is a sub-analysis of the large ‘Mindfetalness’ and it is stated within the paper that there was an analysis between the % of women who accessed healthcare from the Mindfetalness group and routine care, and birth outcomes are presented between the 2 groups, however access to healthcare is not presented, although it is mentioned that Mindfetalness increased the access to

care. I think it would be useful to present the differences in access to health care by background groups as this would help provide insight into whether Mindfetalness is equally effective amongst women with different socio-demographic characteristics, in particular for migrant women.

To comply: We have added a Supplementary Table (1) for an overview of access to care according to background factors divided into Mindfetalness or routine care with absolute differences between the groups. We have referred to the table in the results section in page 10, line 221.

Reviewer: 2
Prof. Shigeki Matsubara, Jichi Medical School

Comments to the Author:

To authors,

The data is interesting and important. I have some advice to improve the paper quality.

1. Introduction: I believe that Mindfetalness is your own contrivance (procedures that your team established/proposed), right? Then, first, please state this meaning (you established the one). Second, shortly state what Mindfetalness is like: I believe that the followings (15 minutes, no counts, subjective self-evaluation, etc.) may be the keys of this procedure/methods.

To comply: We have rewritten the introduction section and for clarity we have added information as suggested, in page 4, lines 104-114:

“Raising maternal awareness of fetal movements by using the Mindfetalness method may be a way forward for women to become familiar with the unborn baby’s movement pattern. Compared to counting methods (ref), Mindfetalness instruct the women to observe the variation of movements. Practicing the method includes to lie down for 15 minutes a day from 28 weeks’ gestation, when the baby is awake and to focus on the character, strength and frequency of the unborn baby’s movements, without counting each movement. Mindfetalness was invented in 2012 (ref) and has been evaluated by women and midwives in two studies (2 ref). Firstly, the women tested Mindfetalness once, and count-to-ten once. The women appreciated both methods but preferred Mindfetalness if they had to choose. Secondly, information about Mindfetalness were distributed to 104 pregnant women in maternity care and the majority liked the method and 75 percent practiced the method. Additionally...”

2. Introduction: I consider that you think that Mindfetalness is “better” than DFM-measurement hitherto reported (there have been many), and thus you did this study. Then, please shortly state in what point Mindfetalness is preferable (superior) to other methods previously reported. If this is one step to progress/widen Mindfetalness, then, “Mindfetalness is very good” is a very important message for the readers.

Answer: See answer to previous question (2).

3. Roughly speaking, this study demonstrated TWO kinds of things at the same time. First, women using Mindfetalness is more likely to visit healthcare-providers than its counterparts. Second, what kind of women is less likely to visit them?: they were: i) aged women, ii) less educated, and iii) non-Swedish women. Then, the relation between the first and second, what? For example, of above three factors (less-likely-visit fraction), Mindfetalness affected what? I mean: “the data showed (or we believe) that Mindfetalness reduced (or may have reduced) the fraction of women of no-visit of i), or ii), or iii), what”. Its clarification may be useful to “improve” Mindfetalness system. If you have no data or no idea regarding this, please state this in the limitation section. I mean that “if you had used conventional count-ten”, i)ii)iii) may have become different.

To comply: We have added information in methodological considerations in page 13, lines 327-334:

“We can only study predictors for seeking care for decreased fetal movements and less likely to seek care were i) aged women, ii) less educated, and iii) non-Swedish women. In the Mindfetalness-group, effects of the intervention affect the results. Women using Mindfetalness were more likely to visit healthcare-providers due to decreased fetal movements than women in the control-group. We believe that Mindfetalness may have reduced the fraction of women of no-visit of

i) aged women, ii) less educated, and iii) non-Swedish women. However, we cannot, in this study draw conclusions that we would not have de same results if we had used conventional count-ten in our study.”

4. The endpoint of DFM should be “to reduce the stillbirth without too much overload on both women and care-givers”. Stillbirth is a very rare phenomenon and thus comparing stillbirth rate in this study is impossible: I do understand. You state the meaning at the last of the text. However, since this journal is not OBYN specific, I believe that this should be definitely mentioned for the ordinary readers. This study did not show the very point; stillbirth reduction.

To comply: We have added information in the introduction section as suggested, in page 5, lines 125-128:

“The ambition to encourage women to seek care if they have concerns of their unborn baby’s fetal movements pattern is to reduce the stillbirth rate without too much overload on both women and caregivers. However, stillbirth is a very rare phenomenon and thus comparing stillbirth rate in this study is impossible.”

5. Also, for educational meaning, instead of count-to ten and other conventional methods (counting method), subjective movement count (like Mindfetalness) may be better to pick-up jeopardized fetus, if you consider so.

Answer: We agree and have added information about this in introduction section in page 4, lines 104-109:

“Raising maternal awareness of fetal movements by using the Mindfetalness method may be a way forward for women to become familiar with the unborn baby’s movement pattern. Compared to counting methods (ref), Mindfetalness instruct the women to observe the variation of movements. Practicing the method includes to lie down for 15 minutes a day from 28 weeks’ gestation, when the baby is awake and to focus on the character, strength and frequency of the unborn baby’s movements, without counting each movement.

6. I made several advices and when you agree with me and decide to incorporate them, please do not expand the paper volume. In my opinion, the present paper is already too long to describe this simple data.

Answer: Thank you, we have made several changes in the manuscript according to reviewers and editor’s suggestion. The first submission the manuscript included 3265 words (from abstract to discussion) and the new version includes 3558 words, i.e. only 293 extra words.

VERSION 2 – REVIEW

REVIEWER	Stacey, Tomasina University of Huddersfield, Department of Nursing and Midwifery
REVIEW RETURNED	26-May-2021

GENERAL COMMENTS	I feel that the authors have generally responded appropriately to the suggestions/recommendations of the reviewers/editors. However a number of editing issues appear to have arisen and need addressing, for instance lines: 42 remove 'in' 44 would better read: ...for women to use daily to become familiar with their unborn baby’s ... 107: add 'fetal' before movements 110-4: I don't think these sentences are clear or add much to the paper 137 'in which' rather than 'in where'
---

	162-4 is a repetition from earlier 206 lesser rather than lower 213 lesser rather than lower 310 likely rather than likeable 329-30 I am not sure what this sentence means, needs rephrasing? Women 35 and over rather than 'aged women' 330-1: this sentence should be removed 333-334 The sentence should removed: 334-6 May read better as: it is unclear from this study whether we would have seen the same results if we had used a conventional 'count to ten' approach to maternal awareness of fetal movements.
--	--

REVIEWER	Matsubara, Shigeki Jichi Medical School, Department of Obstetrics and Gynecology
REVIEW RETURNED	10-May-2021

GENERAL COMMENTS	To authors, The authors faithfully reacted the reviewers' comments, of which incorporation into the version markedly improved the paper quality. Non-Sweden-born women and lower-educated women had less likely to visit the clinic in perceived decreased fetal movements (DFM). Straightforwardly speaking, this is all that this study showed. Some may feel that this data is not significant (scientifically/medically not so meaningful); however, I believe that this is a fundamental step to reduce the stillbirth. Researchers of this issue well understand that "a single study never indicates/identifies how to reduce the stillbirth from the viewpoint of DFM at once and at one time". I wish that the authors may/will continue their effort to identify the methods that actually reduce the stillbirth rate.
--

VERSION 2 – AUTHOR RESPONSE

Reviewer: 1

I feel that the authors have generally responded appropriately to the suggestions/recommendations of the reviewers/editors. However a number of editing issues appear to have arisen and need addressing, for instance lines:

Answer: Thank you!

42 remove 'in'

To comply: We have removed the word in page 1, line 43.

44 would better read: ...for women to use daily to become familiar with their unborn baby's ...

To comply: We have removed the word "the" in page 1, line 45.

107: add 'fetal' before movements

To comply: We have added the word "fetal" in page 4, line 99.

110-4: I don't think these sentences are clear or add much to the paper

To comply: We have removed the sentences in page 4, lines 102-106.

137 'in which' rather than 'in where'

To comply: We have changed the word in page 5, line 128 to "in which"

162-4 is a repetition from earlier

To comply: We have removed the text in pages 5-6, lines 149-151.

206 lesser rather than lower

To comply: We have changed the word in page 8, line 185 to "lesser"

213 lesser rather than lower

To comply: We have changed the word in page 9, line 191 to "lesser"

310 likely rather than likeable

To comply: We have changed the word in page 12, line 286 to "likely"

329-30 I am not sure what this sentence means, needs rephrasing? Women 35 and over rather than 'aged women'

To comply: We have changed the sentence in page 13, line 305-306 to:

"Less likely to seek care were i) women 35 or over..."

330-1: this sentence should be removed

To comply: We have removed the sentence in page 13, lines 306-307.

333-334 The sentence should be removed:

To comply: We have removed the sentence in page 13, lines 309-310.

334-6 May read better as: it is unclear from this study whether we would have seen the same results if we had used a conventional 'count to ten' approach to maternal awareness of fetal movements.

To comply: We have removed the sentence in page 13, lines 310-314 and changed to:

"It is unclear from this study whether we would have seen the same results if we had used a conventional 'count to ten' approach to maternal awareness of fetal movements."

Reviewer: 2

The authors faithfully reacted to the reviewers' comments, of which incorporation into the version markedly improved the paper quality. Non-Sweden-born women and lower-educated women had less likely to visit the clinic in perceived decreased fetal movements (DFM). Straightforwardly speaking, this is all that this study showed. Some may feel that this data is not significant (scientifically/medically not so meaningful); however, I believe that this is a fundamental step to reduce the stillbirth.

Researchers of this issue will understand that "a single study never indicates/identifies how to reduce the stillbirth from the viewpoint of DFM at once and at one time". I wish that the authors may/will continue their effort to identify the methods that actually reduce the stillbirth rate.

Answer: Thank you!